# Effect of Escitalopram on the Number of DCX-Positive Cells and NMUR2 Receptor Expression in the Rat Hippocampus under the Condition of NPSR Receptor Blockade

**DOI:** 10.3390/ph15050631

**Published:** 2022-05-20

**Authors:** Aneta Piwowarczyk-Nowak, Artur Pałasz, Aleksandra Suszka-Świtek, Iwona Błaszczyk, Katarzyna Bogus, Barbara Łasut-Szyszka, Marek Krzystanek, John J. Worthington

**Affiliations:** 1Department of Anatomy, Faculty of Medical Sciences in Katowice, Medical University of Silesia, ul. Medyków 18, 40-752 Katowice, Poland; anowak@sum.edu.pl; 2Department of Histology, Faculty of Medical Sciences in Katowice, Medical University of Silesia, ul. Medyków 18, 40-752 Katowice, Poland; aswitek@sum.edu.pl (A.S.-Ś.); ablaszczyk@sum.edu.pl (I.B.); kbogus@sum.edu.pl (K.B.); 3Center for Translational Research and Molecular Biology of Cancer, Maria Skłodowska-Curie National Research Institute of Oncology, Gliwice Branch, 44-102 Gliwice, Poland; barbara.lasut@gmail.com; 4Clinic of Psychiatric Rehabilitation, Department of Psychiatry and Psychotherapy, Faculty of Medical Sciences in Katowice, Medical University of Silesia, ul. Ziolowa 45/47, 40-635 Katowice, Poland; krzystanekmarek@gmail.com; 5Division of Biomedical and Life Sciences, Faculty of Health and Medicine, Lancaster University, Lancaster LA1 4YG, UK; j.j.worthington@lancaster.ac.uk

**Keywords:** escitalopram, neuropeptide S, hippocampus, neuromedin U, NPSR, NMUR2

## Abstract

Background: Neuropeptide S (NPS) is a multifunctional regulatory factor that exhibits a potent anxiolytic activity in animal models. However, there are no reports dealing with the potential molecular interactions between the activity of selective serotonin reuptake inhibitors (SSRIs) and NPS signaling, especially in the context of adult neurogenesis and the expression of noncanonical stress-related neuropeptides such as neuromedin U (NMU). The present work therefore focused on immunoexpression of neuromedin U receptor 2 (NMUR2) and doublecortin (DCX) in the rat hippocampus after acute treatment with escitalopram and in combination with selective neuropeptide S receptor (NPSR) blockade. Methods: Studies were carried out on adult, male Sprague-Dawley rats that were divided into five groups: animals injected with saline (control) and experimental individuals treated with escitalopram (at single dose 10 mg/kg daily), escitalopram + SHA-68, a selective NPSR antagonist (at single dose 40 mg/kg), SHA-68 alone, and corresponding vehicle control. All animals were sacrificed under halothane anaesthesia. The whole hippocampi were quickly excised, fixed, and finally sliced for general qualitative immunohistochemical assessment of the NPSR and NMUR2 expression. The number of immature neurons was enumerated using immunofluorescent detection of doublecortin (DCX) expression within the subgranular zone (SGZ). Results: Acute escitalopram administration affects the number of DCX and NMUR2-expressing cells in the adult rat hippocampus. A decreased number of DCX-expressing neuroblasts after treatment with escitalopram was augmented by SHA-68 coadministration. Conclusions: Early pharmacological effects of escitalopram may be at least partly connected with local NPSR-related alterations of neuroblast maturation in the rat hippocampus. Escitalopram may affect neuropeptide and DCX-expression starting even from the first dose. Adult neurogenesis may be regulated via paracrine neuropeptide S and NMU-related signaling.

## 1. Introduction

Pharmacology of antidepressant drugs is an important area of contemporary neuroscience and clinical psychiatry. Dynamic approaches of neurochemical studies have resulted in several reports that suggest a distinct involvement of several neuropeptides in the origin of the anxiety responses [1,2,3]. Hence, a hypothesis assuming that pharmacological effects of some antidepressants may be triggered alternatively via modulation of peptidergic signaling seems to be reasonable. An experimental model designed to verify this assumption may also help to understand endogenous mechanisms of depression and to outline potential perspectives in the field of novel therapeutic strategies aimed at modulation of neuropeptide action in the brain.

Adult neurogenesis is considered to be important in the processes of memory, learning, and neural plasticity. In the adult mammalian brain, neural stem cell niches are located in the subgranular zone (SGZ) of the hippocampal dentate gyrus, the subventricular zone (SVZ) of the lateral ventricles, and in some subependymal regions of the hypothalamus. Adult neurogenesis remains an intriguing topic in contemporary neuroscience; however, there is a lack of convincing evidence that would confirm the presence of stable and physiologically relevant long-term neural proliferation in the adult human brain [4,5,6].

Neuropeptides such as neuropeptide Y (NPY), vasoactive intestinal peptide (VIP), and galanin have emerged as important mediators for signalling local and extrinsic interneuronal activity to subgranular zone precursors [7].

A number of recently formulated hypotheses suggest that impaired adult neurogenesis is related to the pathogenesis of mental disorders, e.g., depression, and schizophrenia. Neurogenesis, neuroprotection, and cell death are therefore related to the mechanism of action of antidepressants [8,9,10]. Moreover, several studies also confirm the possibility of antipsychotics-stimulated neurogenesis, mediated via multiple molecular mechanisms [11,12]. The mechanisms involved in neurogenesis caused by second generation antipsychotics are not fully understood, and getting to know them may help in developing more effective strategies for schizophrenia treatment.

Escitalopram is an S-enantiomer of citalopram, a selective serotonin reuptake inhibitor (SSRI) with beneficial pharmacological properties and a satisfactory tolerance profile. Escitalopram has a minimal affinity to serotonin, dopamine, and cholinergic receptors, which highly reduce the range of its potential side effects. SSRI-related changes in energy homeostasis and weight gain are often clinically observed [13,14]; however, little is known about stress-related peptidergic neuronal pathways, which could be additional targets for these medications. Although SSRIs often exhibit anxiolytic and sedative properties, their potential direct or indirect effects on stress-related neuropeptides signalling are hitherto completely unknown.

Recently, several novel stress and anxiety-related neuropeptides have been discovered in the brain. Almost all of them exposed both the unique properties and multidirectional activities at the level of numerous neural pathways. Against this background, newly identified modulators of stress responses neuromedin U (NMU) and neuropeptide S (NPS) seem to be specifically worth investigating. Neuropeptide S (NPS), as a product of 89-amino acid propeptide conversion, is a ligand of the G-coupled receptor (NPSR) formerly known as GPR 154 [15]. NPSR stimulation causes calcium release to the neuroplasm, increases cAMP levels, and probably phosphorylates protein kinase MAPK resulting in the activation of neurons [16,17]. In rodents, a limited population of NPS-expressing glutamatergic neurons is located mainly in the brainstem, hypothalamus, and amygdala [18,19]. The NPS receptor is in turn widely distributed in the rat brain, especially in the hypothalamus, amygdala, hippocampus, olfactory bulb, as well as in some thalamic and cortical regions [20]. NPS is a neuromodulator with a wide spectrum of regulatory activity in the brain; it exposes anxiolytic action, stabilizes wakefulness, regulates food intake, and plays a role in the mechanisms of addiction [15,21,22]. From a neuropsychiatric viewpoint, an anchoring of NPS in the fear-related neural pathways seem to be particularly important. Central NPS administration causes a potent anxiolytic effect in rats connected with elevated dopamine release in the prefrontal cortex but not with modulation of serotoninergic transmission [23,24]. Interestingly, several polymorphisms in the human NPSR gene may potentially increase a risk of panic anxiety episodes [25]. NPS secretion within particular brain structures is probably triggered by exposition to stress [26]. To date, very little is known about the neurochemistry of NPS signalling in the human brain, with a population of NPS mRNA-expressing neurons so far only being found in the pontine gray matter [27].

Neuromedin U (NMU) is an anorexigenic 25-amino acid peptide involved in the regulation of numerous neurophysiological processes [28,29]. In the rat brain, the highest, albeit dispersed, level of NMU immunoreactivity is found in the nucleus accumbens (NAc), hypothalamus, septum, amygdala, globus pallidus, and brainstem [30]. In the human brain, NMU precursor protein was identified in the hypothalamus, NAc, thalamus, locus coeruleus (LC), cingulate, and medial frontal gyri [31]. In comparison, NMU mRNA expression was detected in the rat hypothalamus as well as in the brainstem nuclei [32]. Two types of metabotropic NMU receptors are currently known: NMUR1 and NMUR2, coupled with Gq and Gi/0 proteins, respectively [28]. NMUR1 is present almost exclusively in the peripheral tissues, while in contrast, NMUR2 is expressed predominantly in the CNS, especially in the hypothalamus, thalamus, hippocampus, substantia nigra, and brainstem [33].

The present study aims to shed light on this area by determining if and how acute treatment with escitalopram influences the expression of NMUR receptor 2 (NMUR2) and number of DCX-positive neuroblasts in the adult rat hippocampus. We hypothesize that the potential effect of escitalopram on newborn neurons’ maturation in the SGZ may possibly be related to local modulation of neuropeptide S signalling. The second purpose of the study was also to investigate the possible effect of NPS transmission blockade with SHA-68, a selective NPSR antagonist, on SSRI-related changes in NMUR2 in the context of adult neurogenesis.

## 2. Results

The number of DCX-expressing cells (Figure 1 and Figure 2) was significantly lower in the groups of animals exposed to escitalopram, escitalopram + SHA-68, and SHA-68 compared to the control (saline) group (F4,20 = 31.7; *p* < 0.001). Furthermore, the number of immunopositive neuroblasts was noticeably decreased in the escitalopram + SHA68 group compared to the escitalopram group (F4,20 = 31.7; *p* = 0.015). We also found that the number of DCX-positive cells in animals exposed to SHA-68 solvent (vehiculum) was also reduced (F4,20 = 31.7; *p* = 0.035). Upon qualitative and descriptive morphological assessment of the whole hippocampus of all examined animals, a dense network of NMUR2-expressing fibres was mainly found in the CA1 area (Figure 3). Dendritic trees penetrating stratum radiatum exhibited an abundant NMUR2 immunoreactivity in all experimental groups (Figure 3a,d,g,j). A decreased density of NMUR2-expressing fibres in the SGZ was recorded in both escitalopram and escitalopram + SHA68 groups compared to controls and vehiculum (Figure 3d–i). Noteworthy, an aggregation of intensively NMUR2 positive round or oval cells was also observed in the retrosplenial cortex (Figure 3c,f,i,j). In rats treated with escitalopram + SHA-68, a distinct number of perikarya with their proximal processes seemed to manifest less dense NMUR2 immunostaining (Figure 3h) when compared to controls (Figure 3i). Numerous NPSR-expressing cells were widely distributed in whole CA1, CA3, and dentate gyrus, where most of them exposed high fluorescence intensity (Figure 4). Nevertheless, any significant drug treatment-related quantitative changes in both neuropeptide immunoexpression (e.g., number of positive cells) were not found. 

## 3. Discussion

Neuropsychiatric medications may distinctly affect canonical adult neurogenesis in animal models [9,34,35]. Most of them can cross the blood-brain barrier and modulate some signalling pathways within neural stem cell niches. Several studies reported that SSRI may exhibit differing, age- and sex-related effects on adult neurogenesis in animal models [36,37,38]. Antidepressants from diverse pharmacological groups, e.g., fluoxetine (selective serotonin reuptake inhibitor, SSRI), venlafaxine (selective noradrenaline reuptake inhibitor, NSRI), and tranylcypromine (monoaminooxidase inhibitors, IMAO) act as relatively potent stimulators of adult hippocampal neurogenesis [8,9,10,39,40]. The stimulatory effect of aforementioned antidepressants on NSCs proliferation and subsequent activation of DCX-expressing neuroblasts is the aftermath of neuronal MAPK/ERK i Wnt/GSK-3 pathways modulation [41,42,43]. Nevertheless, our results seem to surprisingly be in line with a contradictory study showing that extended treatment with other SSRI antidepressant fluoxetine decreased the number of immature DCX-expressing neurons in the baboon hippocampus [44]. In contrast to this work, however, we used a single dose of escitalopram instead of chronic treatment, and the experiment was carried out on rat brains where differentiation of newborn neurons is significantly faster than in primates [45]. It should not be excluded that even a single dose of escitalopram as well as other SSRIs such as fluoxetine may accelerate the maturation of neuroblasts as has been observed in mice hippocampus [8,46]. Such an accelerated maturation might result in a more rapid loss of DCX expression and thus lower numbers of DCX-positive cells. As these more mature neurons no longer express DCX, increased maturation may not have been reflected in our analysis of DCX-positive cells. The mechanism of this effect is not known. It should be noted that some SSRIs including fluoxetine may modulate several aspects of brain neurochemistry in a serotonin-independent manner, e.g., via inhibition of cellular Wnt signalling [47] and voltage-gated potassium channels closing [48]. Interestingly, only acute but not chronic fluoxetine administration affects inhibitory synapse formation [49] while extended SSRI treatment has been shown to upregulate adult neurogenesis and neuroplasticity through increased brain-derived neurotrophic factor (BDNF) signalling [50]. It seems to stay in accordance with a hypothesis suggesting that antidepressants may also act starting at the first dose, and some observable neurochemical changes may appear within hours of administering a single dose. A differential psychopharmacological or molecular effect can possibly be present following acute rather than long-term SSRI administration before detectable behavioural changes occur [51,52]. Moreover, escitalopram + SHA-68 coadministration as well as injection of only SHA-68 caused a further significant reduction in the number of DCX-immunopositive cells (Figure 2), suggesting that NPS signalling may play an unknown role in the process of neural precursors differentiation/maturation in the rat hippocampus. A distinct number of NPSR expressing cells in the dentate gyrus (Figure 4) may support this hypothesis; however, a possible coexpression of NPSR with DCX and other neurogenic markers such as Sox-2, TUC-4, and Musashi1 in the SVZ should definitely be examined. On the other hand, DCX protein seems to be particularly sensitive to stress in some mammals, e.g., its hippocampal expression drops rapidly upon 30 min after capture in wild-caught microchiropteran bats [53]. An acute i.p. administration of escitalopram, SHA-68, or even DMSO-containing solvent medium can possibly be considered a stressogenic stimulus that affects neural progenitor differentiation; however, saline injection does not reduce the number of DCX-expressing cells. SHA-68 is a polycyclic, fluorinated compound insoluble in aqueous solutions. A mixture of organic solvents based on Cremophor and dimethylsulfoxide (DMSO) must therefore be used to prepare a stable and homogenous SHA-68 solution [54]. It should be taken into account that DMSO itself is not a physiologically inert agent; its administration may increase functional expression of several receptors, e.g., NMUR or bradykinin in cell cultures [55,56,57]. DMSO is considered to affect motor activity and to modulate the sleep architecture in rats; however its effect on adult neurogenesis is not yet known [58]. Moreover, several reports indicate that DMSO may modulate synaptic transmission at the level of hippocampal circuits [59]. DMSO was also considered a proapoptotic agent that distinctly stimulate cell death in an in vitro rat hippocampal culture preparation [60]. On the other hand, low concentrations (1%) of DMSO reduced oligodedrogenesis but stimulate astrogenesis in neural stem and progenitor cells cultures (NSPC) from the adult hippocampus [61]. Noteworthy, even very low concentrations (0.05%) of DMSO decreased the input resistance of hippocampal neurons with concomitantly reduced excitability [62]. Given all aforementioned considerations, it should be highlighted that the effect of SHA-68 cannot be precisely evaluated because it is not possible to determine the extent to which the DMSO contributed to this effect. An additional control group with the SHA-68 solvent (vehiculum; VC) was therefore established in our study. Nevertheless, some new physiologically inert but DMSO-free SHA-68 solvents have to be urgently designed. Despite all imperfections, our initial finding touched upon some structural issues that may potentially be interesting from the neuropharmacological viewpoint and so far have not been investigated. It should be perceived as a starting point to a broader examination of novel mechanisms of antidepressant action.

The hippocampus has been known to be an important part of the neuromedin U signaling system, which receives its NMU projections from the lateral hypothalamus mainly. Indeed, upon examining the hippocampal area, a distinct number of neurons and glial cells exhibited NMUR2 immunoreactivity. We have also shown a distinct expression of NMUR2 in the dentate gyrus of control rats, particularly abundant in the subgranular zone (Figure 3). These may cautiously suggest a so far understudied relationship between neuropeptide S and neuromedin U signaling and hippocampal adult neurogenesis. However, further studies are definitely required to support this possible regulatory interaction. Noteworthy pretreatment with NMU administration prevented the LPS-related glial cells death in vitro; however, NMU exposed no effect on hippocampal neuronal degeneration induced directly by interleukin-1beta administration. On the other hand, NMU increased the BDNF level in rat hippocampus that may support neuronal viability in this region [63]. Stimulation of brain NMUR2 has been found to modulate anxiety-like behaviour and trigger stress-related molecular events by CRH exocytosis [64,65]. NMU-23 has been shown to have antidepressant-like behavioural effects in mice [64]. On the other hand, local NMU signaling circuits in nucleus accumbens acting via NMUR2 may also reduce reward responses induced by several psychoactive substances and ethanol [66,67]. Interestingly, numerous neurons in the retrosplenial cortex (RSC) exhibited a distinct NMUR2 immunoexpression. The RSC is reciprocally connected with the hippocampus, suggesting that its functionally plastic neuron region not only regulates sensory input to the hippocampal circuit, but may also be considered a crucial site of information storage. Recent evidence reports that RSC acting as an important part of the so-called “where/when” cortical pathway may be essential for consolidating both spatial and contextual memories [68]. It should not be excluded that NMUR2-related signalling may play a known role in the aforementioned process. The mechanism of neuropeptides action at the level of stem cell niches is in general still not clear, e.g., short exposure to neuropeptide Y (NPY), a potent stimulator of subventricular neurogenesis increased the nuclear level of phosphorylated form of extracellular signal-regulated kinase 1/2, starting cell proliferation. More extended 6 h-long NPY administration amplified the phosphorylated form of c-Jun-NH(2)-terminal kinase signal in growing axons, consistent with axonogenesis [69]. In the mouse hippocampus, the proliferative effect of NPY is mediated by the Y1 but not the Y2 receptor. NPY-induced neural proliferation in SGZ is abolished by Y1 antagonist administration. The same effect was observed in Y1(-/-) mice [70]. In the present study we have shown for the first time that early pharmacological effects of escitalopram may be at least partly connected with local NPSR-related alterations of NMUR2 signalling and neuroblast maturation in the rat hippocampus.

Nevertheless, we have to point out several limitations of the study, which have to be taken into account. Firstly, quantitative analysis of immunohistochemical reactions for NPSR and NMUR2 was not carried out and gene expressions were not measured. Secondly, there was also relatively small number of rats; however, we have applied a set of appropriate statistical methods. Behavioural tests were not performed, but this will be provided in our ongoing research project. To conclude, it is definitely worth considering expanding the experiment to complement our initial report, e.g., it should be supplemented by the analysis of NMUR2 and NPSR coexpression with some markers of adult neurogenesis.

## 4. Materials and Methods

### 4.1. Animals 

The studies were carried out on adult (2–3 months old, 180–210 g) male Sprague-Dawley rats from Medical University of Silesia Experimental Centre housed at 220 °C with a regular 12/12 light-darkness cycle with access to standard Murigran chow and water ad libitum. All experimental procedures were approved by the local bioethics committee at the Medical University of Silesia (agreement no 36/2012, dated 18 April 2012).

### 4.2. Drug Administration 

Five groups of rats (*n* = 10) have received, respectively, control vehicle (physiological salt, 0.25 mL), escitalopram (10 mg/kg), escitalopram (10 mg/kg) + SHA-68 (40 mg/kg), SHA-68 (40 mg/kg), and SHA-68 solvent (0.25 mL) by a single intraperitoneal injection. Escitalopram + SHA-68 have been administered in the form of two independent injections (drug and inhibitor). Above mentioned non-toxic doses of drugs were established on the basis of pharmacological standards developed in preclinical studies focused on the antidepressant actions on the neuropeptide at the level of peptidergic signalling. Escitalopram in the form of powder was dissolved in saline solution (NaCl 0.9%), crystalline SHA-68 (NPSR receptor antagonist) in the PBS containing 10% Cremophor EL, and 10% DMSO and injected i.p.

### 4.3. Brain Tissue Collection

Twenty-four hours after the last drug administration, animals were anaesthetized and perfused with 4% paraformaldehyde PBS (pH 7.2–7.4). The hippocampi with an ipsilateral retrosplenial cortex were quickly excised, postfixed, dehydrated, embedded in paraffin, and finally sectioned on the microtome (Leica Microsystems, Wetzlar, Germany) in the coronal plane (−2.00 to −2.80 mm from bregma) at 7 μm thick slices, according to Paxinos & Watson’s The Rat Brain in Stereotaxic Coordinates (2007) [71].

### 4.4. Immunohistochemistry and Immunofluorescence

For immunohistochemical assay of neuroblast distribution in the SGZ, after blocking with 0.1% Triton X-100 (Sigma, T-7878, Darmstadt, Germany) and 10% serum, sections were incubated overnight at 40 C with the goat anti-rat DCX monoclonal antibody (Santa Cruz Pharmaceuticals, Dallas, TX, USA, 1:200). Brain sections were also incubated with polyclonal rabbit antibodies against the following rat antigens: neuropeptide S receptor (1:500, Bioss Antibodies, bs-11430R, Woburn, MA, USA) and NMUR2 (1:500, Novus Biologicals, Cambridge, UK, NBP1-02351). Primary antibodies against DCX and NPSR were followed by fluorochrome-conjugated secondary antibody: goat anti-rabbit FITC (1:200, Abcam, Cambridge, UK) for 1 h at 40 C. Overnight incubation with primary antibodies against NMUR2 were followed by administration of biotynylated anti-goat/anti-rabbit secondary antibody (1:200), and then an avidin-biotin-horseradish peroxidase complex (Vectastain ABC kit, Vector Labs, San Francisco, CA, USA). 3,3′-diaminobenzidine (DAB) was used to complete the reaction and visualize receptors expressing neurons. Sections incubated with rabbit/mouse IgG instead of primary antibody were used as negative controls in order to check specificity of primary antibodies. Finally, sections were mounted on glass slides with DAPI-containing medium or DPX and coverslipped. All images were captured and analyzed with Nikon Coolpix optic systems and processed using Image ProPlus software (Media Cybernetics, Rockville, MD, USA). The same planes of the brain were chosen from each set of slides. For the calculation of NMUR2 and DCX-expressing cells, 5 slices (every fifth one from the series) per rat for each hippocampal slice were used. Anatomically comparable sections were analyzed and immunopositive cells were counted using ImageJ 1.43 u software. We counted the total number of DCX-expressing cells in the subgranular zone for each rat (which was the sum of cells from 10 slices) and then divided the results per length of the analyzed dentate gyri (SGZ) to obtain number of immunopositive cells per one millimeter of length. All green fluorescent or brown stained multipolar/polygonal cells with the diameter not smaller than 20 μm and visible nuclei were counted. Exclusion criteria were as follows: cells located outside of the SGZ, perikarya with invisible nuclei, cell size less than 20 μm, yellow or too intense (white) fluorescence, regular round shape without dendrites, or protrusions. The immunopositive processes without their perikarya were also omitted.

### 4.5. Statistical Analysis

Statistical analysis was performed using data analysis software system Statistica (TIBCO Software Inc. 2017, version 13). Data is presented as mean ± SEM. Mean differences between groups were analyzed using ANOVA followed by Tukey’s post hoc test. Differences were considered statistically significant at *p* < 0.05 with two confidence levels (0.01 ≤ *p* < 0.05 and *p* < 0.01).

## 5. Conclusions

Pharmacological effects of escitalopram may be at least partly connected with local neuropeptide S-related alterations of neuroblast maturation in the rat hippocampus. A single injection of escitalopram and NPSR receptor antagonist SHA-68 decrease the number of DCX-expressing immature neurons in the SGZ. Neuromedin U receptor 2 (NMUR2) expression in the SGZ seems to be also reduced; however, further fully quantitative studies are required to confirm this preliminary observation. Escitalopram may act starting even at the first dose and several changes in the DCX-expression may appear within hours of administering an acute dose of the antidepressant. This can indirectly support a recently suggested possibility that adult neurogenesis may be regulated via paracrine neuropeptide S and NMU-related signaling. The preliminary results may cautiously suggest that SSRIs neurochemical activity at the level of hippocampus is not limited to their interactions with serotonin signalling, but also affects some peptidergic pathways. However, further molecular studies are urgently required to confirm this potential regulatory interplay.

## Figures and Tables

**Figure 1 pharmaceuticals-15-00631-f001:**
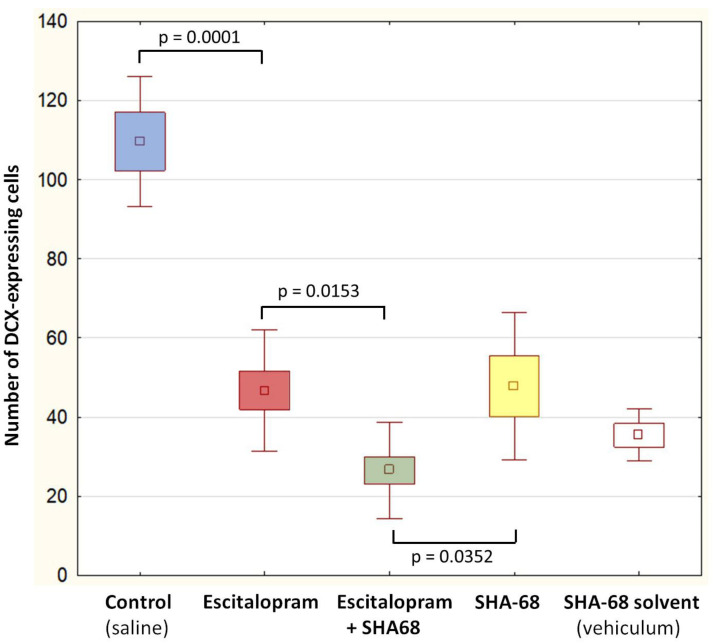
The number of DCX immunopositive cells in the subgranular zone (SGZ). Animals received: 0.25 mL saline—Control, escitalopram (10 mg/kg)—Esc, escitalopram (10 mg/kg) with SHA-68 (NPSR antagonist, 40 mg/kg)—Esc + SHA-68, only SHA-68 (40 mg/kg), SHA-68 solvent (0.25 mL)—Vehicle, by a single intraperitoneal injection. Boxes show mean SEM and SD, *n* = 5. Differences between groups were statistically analyzed using ANOVA followed by Tukey’s post hoc test, and they were considered significant at *p* < 0.05.

**Figure 2 pharmaceuticals-15-00631-f002:**
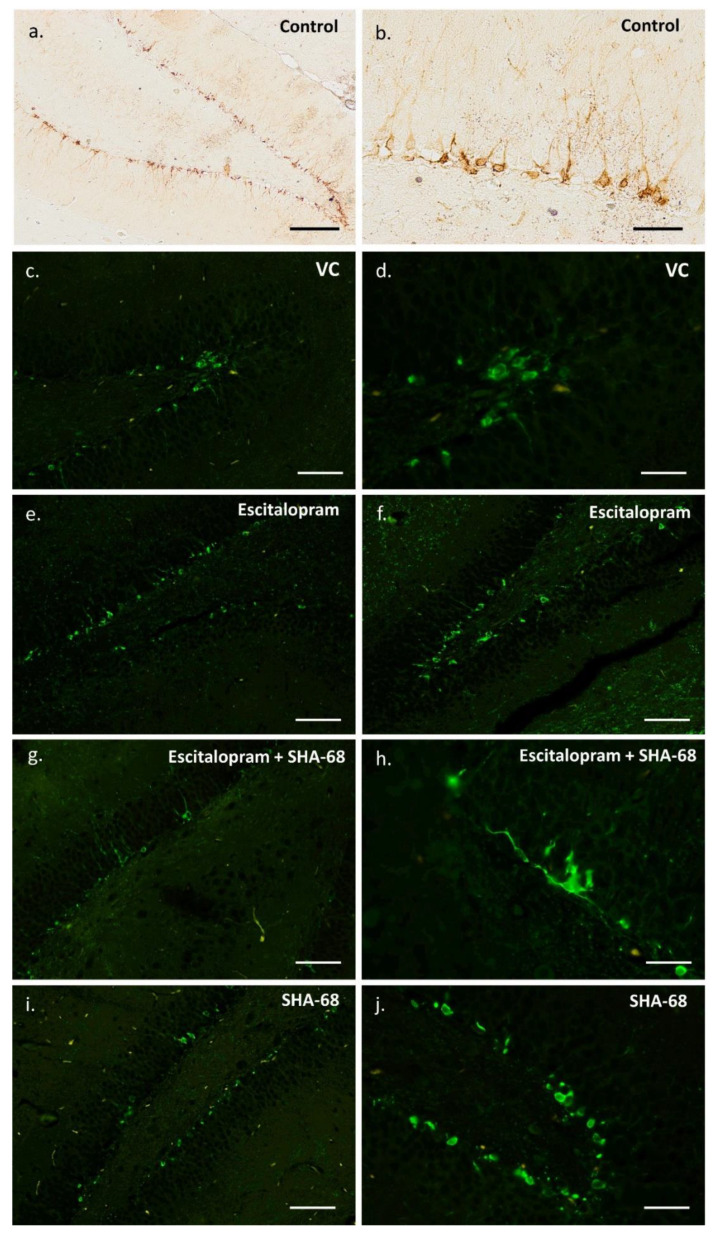
DCX-expressing immature neurons in the neurogenic subgranular zone (SGZ) of control rats (**a**,**b**) and animals after treatment with: escitalopram (**e**,**f**), escitalopram + SHA68 (**g**,**h**), SHA-68 (**i**,**j**) and vehiculum (SHA-68 solvent; (**c**,**d**)). Scale bars: 50 μm (**b**,**d**,**h**,**j**), 100 μm (**a**,**c**,**e**–**g**,**i**). Fluorescence, secondary antibody coupled with FITC. Images captured with Nikon Eclipse Ti microscope.

**Figure 3 pharmaceuticals-15-00631-f003:**
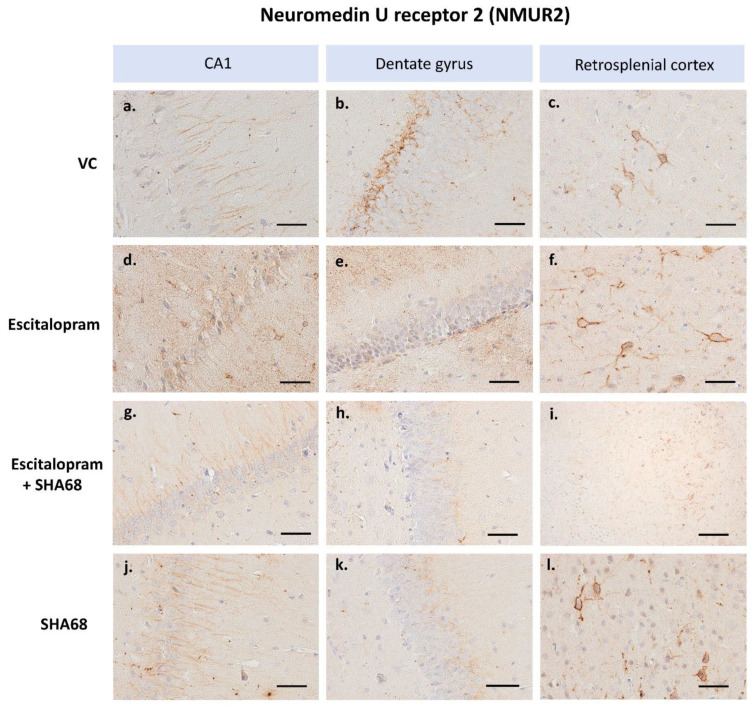
Representative expression of NMUR2 in the hippocampus and retrosplenial cortex (RSC). Immunopositive neurons in the CA1 area, dentate gyrus, and RSC of rats after treatment with: escitalopram (**d**–**f**), escitalopram + SHA68 (**g**–**i**), SHA-68 (**j**–**l**) and vehiculum (SHA-68 solvent; (**a**–**c**)). Immunoperoxidase reaction with DAB staining. Numerous fibres in the hippocampus and perikarya of RSC neurons exhibit a distinct NMUR2 immunoreactivity. Scale bars: 50 μm (**c**,**f**,**l**), 100 μm (**a**,**b**,**d**,**e**,**g**,**h**,**j**,**k**) 200 μm (**i**). Images captured with Nikon Eclipse Ti microscope.

**Figure 4 pharmaceuticals-15-00631-f004:**
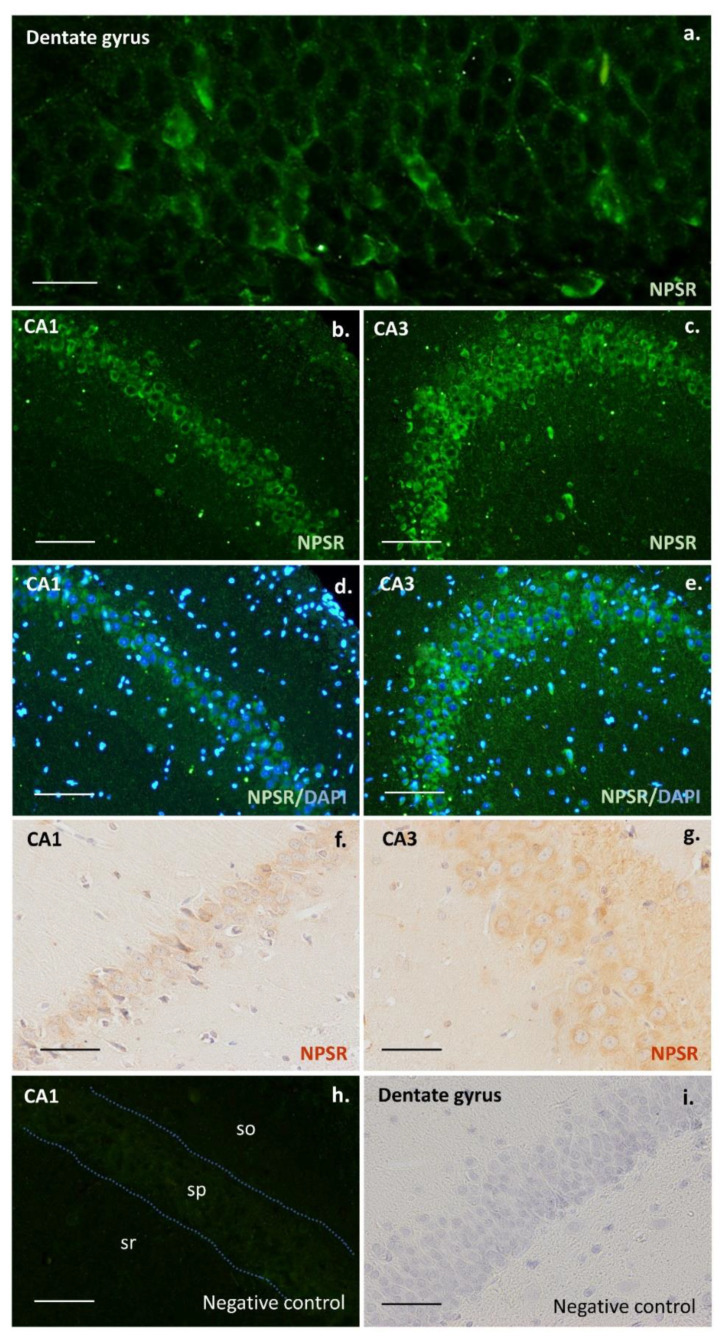
Neuropeptide S (NPSR) receptor-expressing neurons in the rat hippocampus. Immunopositive neurons in the dentate gyrus, CA1, and CA3 areas of control rats. Fluorescence, secondary antibody coupled with FITC (**a**–**e**). Immunoperoxidase reaction with DAB staining (**f**,**g**). Negative controls with omission of primary antibody against rat NPSR (**h**,**i**). Scale bars: 20 μm (**a**), 100 μm (**g**), 200 μm (**b**–**f**,**h**,**i**). Images captured with Nikon Eclipse Ti microscope.

## Data Availability

Not applicable.

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
