# Peer review of "Effect of Escitalopram on the Number of DCX-Positive Cells and NMUR2 Receptor Expression in the Rat Hippocampus under the Condition of NPSR Receptor Blockade"

_pharmaceuticals, 2022, doi:10.3390/ph15050631_

Round 1

Reviewer 1 Report

The paper in principle is interesting and addresses important issue in the field of neuropharmacology. However, there are some major methodological concerns:

- List the exclusion and inclusion criteria for cell counting. Are 7 μm thick slices enough to include the DCX immunoreactive size? This is really important to avoid cofounding results (by including the same stained cells within the counting)

- Could Authors confirm these quantitative data by WB in the DG and CA1?

- Neuropeptide S (NPSR) receptor-expressing neurons in the rat hippocampus: proposed immunofluorescence images are suspicious of autofluorescence, can authors provide more images of higher quality? Nuclei counterstaining would be greatly appreciated.

Author Response

Reviewer 1.

The paper in principle is interesting and addresses important issue in the field of neuropharmacology. However, there are some major methodological concerns:

Thank you for your warm opinion regarding our manuscript. We are very glad you find our article interesting and important.

- List the exclusion and inclusion criteria for cell counting. Are 7 μm thick slices enough to include the DCX immunoreactive size? This is really important to avoid cofounding results (by including the same stained cells within the counting)

Thank you for this comment. Indeed, we did not use confocal microscopy for analysis of thicker slices, however we analyzed every fifth slide from the series to avoid counting the same cells. All green fluorescent or brown stained multipolar/polygonal cells with the diameter not smaller than 20 mm and visible nuclei were counted. The immunopositive processes without their perikarya were omitted.  Exclusion criteria were as follows: cells located outside of the SGZ, perikarya with invisible nuclei, cell size less than 20 mm, yellow or white fluorescence, regular round shape without processes. The aforementioned section has been inserted to the Methods.

- Could Authors confirm these quantitative data by WB in the DG and CA1?

Thank you for this valuable and important suggestion, however due to lack of appropriate tissue samples WB assay can not be carried out. Nevertheless, our studies on NPSR signaling should definitely be expanded. In fact, the experiment will be continued to confirm our assesment by real-time PCR and Western blotting. However, we hope the results presented in our first report  turned out quite interesting and – in our opinion- worth publishing as a short communication. Especially in case when there is no data available dealing with hippocampal neurogenesis and neuropeptide expression after inhibition of neuropeptide S signalling. We have realized well that our initial results are so far imperfect and they should be treated as a new potentially interesting voice in discussion rather than a hypothesis fully supported by experimental data.

- Neuropeptide S (NPSR) receptor-expressing neurons in the rat hippocampus: proposed immunofluorescence images are suspicious of autofluorescence, can authors provide more images of higher quality? Nuclei counterstaining would be greatly appreciated.

Thank you for this important concern. Yes, autofluorescence is considered very common problem that may diminish reliability of the results. To verify the selectivity of NPSR immunofluorescence, peroxidase-DAB reaction has been also performed (new images inserted). Moreover, images depicting negative controls (omission of primary antibody) have been added as well as two micrographs showing nuclei counterstaining with DAPI.

Reviewer 2 Report

The authors of the presented manuscript studied the effect of escitalopram and escitalopram with SHA-68 (blocker of receptor for neuropeptide S) application on the number of DCX-positive immature neurons in the hippocampus.   They also studied the presence of the neuromedin U receptor in this area of the brain by means of immunofluorescence. This is an interesting topic dealing with the possible role of neuropeptide S in neurogenesis in adults. Unfortunately, there are some inaccuracies in the manuscript that need to be clarified.

  1. The title does not match the data obtained. The methodology used to assess the effect of escitalopram on NMUR2 expression is not quantitative. Therefore, in this way, it is not possible to determine whether escitalopram affects the expression of the above mentioned receptor. The declared main aim of the study, assessment of influence of escitalopram treatment on expression of NMUR2 could not be reach by this methodical approach. It is not possible to draw a conclusion from the results obtained as it is written in the manuscript. Real quantitative measurement should be done.
  2. The effect of SHA-68 cannot be evaluated because it is not possible to determine the extent to which the solvent contributed to this effect. This should be communicated much more clearly in the discussion.
  3. Concerning to the figure 1: In the figure legend, there is a sentence which is not related to this figure while it speaks about reference gene. Number of used animals in each group should be mentioned in the legend. Unit on y-axis is missing.
  4. Concerning to figure 2: In the figure legend, there is not mentioned information about scale bar of B.
  5. Concerning to figure 3: In the figure legend, there is probably typing error concerning to scale bars at the end of line 217 a-b instead of a-d.
  6. Concerning to figure 4: In the figure legend, there is mentioned area CA2 but in the figure C is written CA3.

Author Response

Reviewer 2.

The authors of the presented manuscript studied the effect of escitalopram and escitalopram with SHA-68 (blocker of receptor for neuropeptide S) application on the number of DCX-positive immature neurons in the hippocampus.   They also studied the presence of the neuromedin U receptor in this area of the brain by means of immunofluorescence. This is an interesting topic dealing with the possible role of neuropeptide S in neurogenesis in adults. Unfortunately, there are some inaccuracies in the manuscript that need to be clarified.

  1. The title does not match the data obtained. The methodology used to assess the effect of escitalopram on NMUR2 expression is not quantitative. Therefore, in this way, it is not possible to determine whether escitalopram affects the expression of the above mentioned receptor. It is not possible to draw a conclusion from the results obtained as it is written in the manuscript. Real quantitative measurement should be done.

Thank you for your warm opinion regarding our manuscript. We are happy you find our topic interesting.

Indeed, we absolutely agree with this important statement. Because the complete aim of the study  can not be reach by this methodical approach, the title has been changed into the more appropriate form:  “Effect of escitalopram on the number of DCX-positive cells and NMUR2 receptor expression in the rat hippocampus under the condition of NPSR receptor blockade”. A statistically significant effect of escitalopram on the number of DCX-expressing neuroblasts is currently highlighted. Moreover, the new title does not suggest directly that changes in the neuropeptide expression and neurogenesis are related to drug treatment and NPSR inhibition. We realize well that speculations dealing with NMUR2 expression are not supported by our data (due to lack of qualitative assessment), so Discussion and Conclusion sections have been partly reedited to avoid an overinterpretation. We have underlined that our initial results, while imperfect should be treated as a new potentially interesting voice in discussion rather than a hypothesis fully supported by experimental data.

  1. The effect of SHA-68 cannot be evaluated because it is not possible to determine the extent to which the solvent contributed to this effect. This should be communicated much more clearly in the discussion.

Yes, indeed. That is why an additional control group with DMSO solvent has been established in our study. Thank you for this important comment. This problem has been discussed more precisely with a special attention to  potential effects of DMSO on neural structures both in vivo and in vitro.

  1. Concerning to the figure 1: In the figure legend, there is a sentence which is not related to this figure while it speaks about reference gene. Number of used animals in each group should be mentioned in the legend. Unit on y-axis is missing.

Thank you for this comment, the redundant phrase has been removed. The number of animals is currently mentioned and y-axis is described.

  1. Concerning to figure 2: In the figure legend, there is not mentioned information about scale bar of B.

Revised.

  1. Concerning to figure 3: In the figure legend, there is probably typing error concerning to scale bars at the end of line 217 a-b instead of a-d.

Yes, indeed. The figure legend  has been revised.

  1. Concerning to figure 4: In the figure legend, there is mentioned area CA2 but in the figure C is written CA3.

There was a mistake in the caption, it is CA3. It should be also highlighted that Figure 4. has been changed and expanded according to Reviewer 1 suggestions.

Round 2

Reviewer 1 Report

It can be accepted for publication

Author Response

Thank you very much for your interest regarding our article.

Reviewer 2 Report

The manuscript has been sufficiently improved. I have just small comment:

In the figure legend for figure 2 is missing information about scale bar for picture f.

Author Response

Thank you for this comment. The figure caption revised.